# Base-resolution UV footprinting by sequencing reveals distinctive damage signatures for DNA-binding proteins

Kerryn Elliott [1], Vinod Kumar Singh [1], Martin Boström [1] & Erik Larsson [1] ✉

Decades ago, it was shown that proteins binding to DNA can quantitatively alter the formation of DNA damage by UV light. This established the principle of UV footprinting for non-intrusive study of protein-DNA contacts in living cells, albeit at limited scale and precision. Here, we perform deep base-resolution quantification of the principal UV damage lesion, the cyclobutane pyrimidine dimer (CPD), at select human promoter regions using targeted CPD sequencing. Several transcription factors exhibited distinctive and repeatable damage signatures indicative of site occupancy, involving strong (up to 17-fold) position-specific elevations and reductions in CPD formation frequency relative to naked DNA. Positive damage modulation at some ETS transcription factor binding sites coincided at base level with melanoma somatic mutation hotspots. Our work provides proof of concept for the study of protein-DNA interactions at individual loci using light and sequencing, and reveals widespread and potent modulation of UV damage in regulatory regions.

The cyclobutane pyrimidine dimer (CPD) is the principal UV-induced DNA photoproduct, formed by dimerization of neighboring cytosines and thymines following light absorption[1]. Already in the 1980's, it was shown that protein-DNA contacts, due to their effect on DNA structure, can induce base-specific, quantitative changes in CPD formation propensity[2,3]. This established UV light as a footprinting agent, allowing non-intrusive study of protein contacts in genomic DNA inside of living cells[4]. Importantly, CPD formation may be both stimulated and inhibited at different positions and to variable degrees within a single protein binding site, in a manner that depends on the identity of the bound protein[4]. These effects are rooted in the positioning and flexibility of DNA bases: UV photoreactivity may be inhibited when the mobility of particular pyrimidines is restricted by proteins, while damage may be stimulated at bases that are brought into proximity such that CPD formation is favored[4–6]. UV footprinting can thus potentially provide rich information beyond basic DNA accessibility, adding to its appeal compared to commonly used methods[7]. In practice, UV-based probing of DNA has been hampered by the requirement for quantitative

measurements of CPD damage at single-base resolution, thus far achieved at severely limited scale and precision using gel-based assays[8].

More recently, UV lesions have been mapped genome-wide in human UV-exposed cultured cells using sequencing-based methods[6,9–11]. Rather than per-base quantitative data, these methods are designed to give a scattered global view of random damage lesions: while a few outlier bases may exhibit several CPDs[12], the average genomic dipyrimidine (diPy) will not see even a single detection due to the information being spread thinly genome-wide. Genome-wide mapping has still been useful for studying regional effects, revealing for example that chromatin state in human cells has little influence on net CPD damage burden[9]. Furthermore, while genome-wide CPD data are inadequate for probing damage levels at individual genomic positions, summarization over hundreds or thousands of experimentally supported transcription factor (TF) binding sites has shown that different DNA proteins exhibit different average patterns of UV damage modulation[6,9,10,13]. This approach has also supported that elevated CPD formation at TF binding sites, in particular of the ETS family, contributes to widespread formation of non-coding mutation hotspots

[1]Department of Medical Biochemistry and Cell Biology, Institute of Biomedicine, The Sahlgrenska Academy, University of Gothenburg, SE-405 30 Gothenburg, Sweden. ✉e-mail: erik.larsson@gu.se

in melanoma and other skin cancers[6,10,14]. While these results show the potential for using light and sequencing for probing protein-DNA interactions in cells, UV damage modulation patterns at individual genomic loci remain hidden due to the sparsity of current genome-wide CPD mapping data.

## Results

### Digital quantitative UV footprinting of human promoters at base resolution

To gain insights into how DNA-binding proteins in the cellular environment modulate UV damage at specific genomic positions, we devised a protocol to allow quantification of CPD formation propensity at single-base resolution (Fig. 1a). This was achieved by combining an Illumina-based protocol for CPD mapping[10], based on digestion with a CPD-specific endonuclease[15,16], with a sequence capture assay (Capture CPD-seq, Fig. 1b, Methods). Using this approach, we targeted a diverse set of human promoter regions, each being 3-4 kb in size, as well as one 5 kb intergenic region; 21 regions in total (Fig. 1c, Supplementary Table 1). The targeted regions included several genes with well-characterized regulatory sites, including serum-inducible immediate early promoters (such as *EGR1* and *FOS*[17–19]) harboring known serum response elements (SREs) and being regulated by ETS family TFs and serum response factor (SRF) via CArG elements[20], as well as promoters known to be frequently mutated in skin cancers (including *DPH3*, *RPL13A* and *TERT*[6,10,21–25]).

HeLa cells were exposed to high-intensity (1000 J/m²) UVC (254 nm) light and then immediately harvested to avoid influence from DNA repair. Nine independent cellular replicates were assayed, of which three represented normal untreated cells, three were serum-starved and three serum-stimulated (Fig. 1d). Similarly, nine purified (acellular) DNA samples from HeLa were exposed to UVC and profiled to be used as reference, enabling study of differential CPD signals arising due to protein binding events in cells compared to naked DNA. Finally, three non-UV-exposed control samples were included, for a total of 21 samples (Fig. 1d). Read mapping, removal of PCR duplicates and further data processing (Methods) thus resulted in a multidimensional dataset encompassing 18 informative replicates × 21 assayed regions × 3−5 kb of base-resolution CPD count/level data (Supplementary Data 1).

We found that CPD detections occurred primarily at diPy sites (i.e. TT, TC, CT or CC) in the assayed regions as expected, and that no-UV control samples produced few detections (Fig. 1e). In agreement with efficient formation of thymine dimers by UV[26], TT's exhibited the highest CPD formation frequency, and as many as 175 and 165 CPDs were detected on average at each individual TT site in the combined cellular and naked samples, respectively, reflecting deep CPD coverage across the targeted regions (Fig. 1e). The average number of CPD detections per diPy site varied from 50 to 258 between the regions (combined cellular samples), in part reflecting differences in genomic copy number (Fig. 1f, Supplementary Fig. 1).

As an example, the *FOS* promoter region, which was in the lower half in terms of coverage, exhibited on average 69 CPD detections per

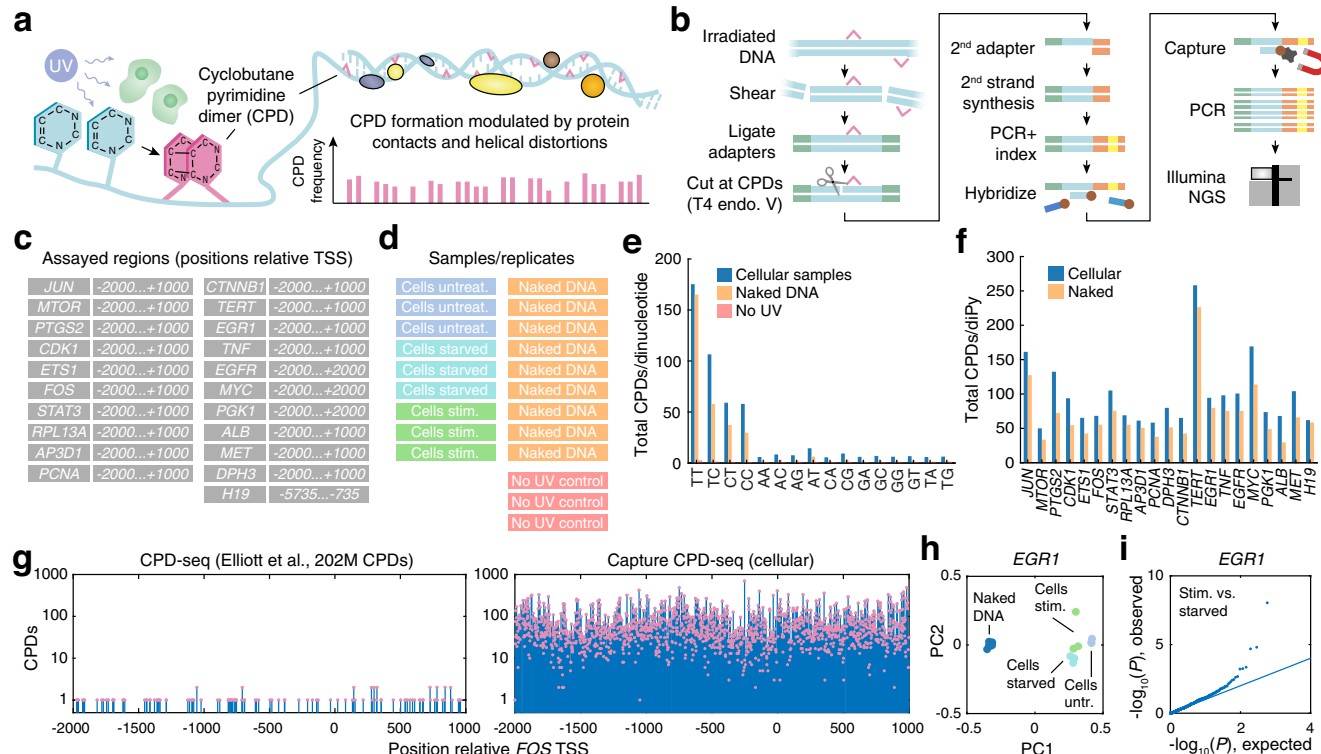

**Fig. 1 | Base-resolution quantitative UV damage footprinting by sequencing.**
**a** Quantification of UV damage (CPD) formation propensity at single-base resolution can reveal changes in local UV damage patterns induced by proteins binding to DNA inside of living cultured cells. **b** Capture CPD-seq enables deep sampling of UV damage formation at regions of interest through enzymatic digestion at CPDs (T4 endonuclease V) combined with hybridization of complementary capture oligonucleotides. **c** Targeted regions (positions are relative gene TSSs). **d** Nine independent cellular samples including serum starved and stimulated and nine naked DNA samples were UV-exposed and assayed, in addition to three non-UV-exposed controls. **e** Number of CPDs detected per dinucleotide in the assayed regions, based on the combined cellular, naked or unexposed samples. **f** Number of CPDs

detected per individual diPy in each of the targeted sequences, based on the combined cellular or naked samples. **g** CPD detections in the *FOS* promoter region in genome-wide non-enriched CPD-seq data[10] (left) compared to Capture CPD-seq (right), illustrating the per-base quantitative nature of the data (total detections in cellular samples shown). **h** PCA analysis of *EGR1* CPD count vectors from the 18 UV-exposed samples reveals that samples cluster based on experimental condition. **i** Quantile-quantile plot (expected vs. observed uncorrected *P*-values, two-sided negative binomial test) showing changes in CPD formation at some positions in the *EGR1* promoter in serum-stimulated compared to starved cells. CPD Cyclobutane pyrimidine dimer, TSS Transcriptome start site, diPy dipyrimidine. Source data are provided as a Source Data file.

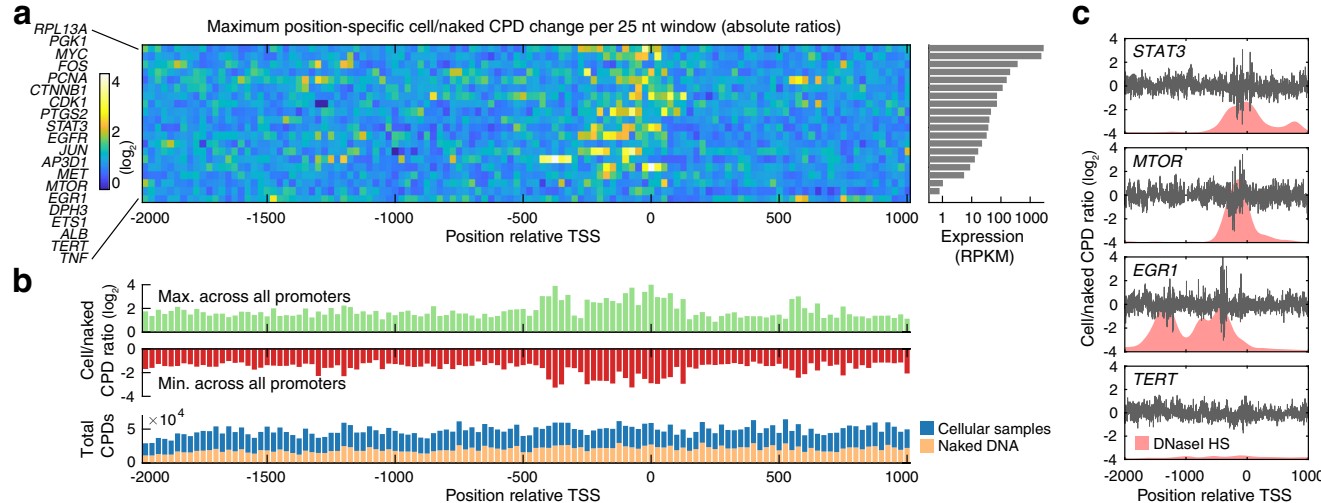

**Fig. 2 | Altered CPD formation immediately upstream of transcription starts in expressed genes. a** Overview of CPD changes across 20 promoter regions. The heatmap shows, for each 25 bp window and each promoter, the strongest base-specific change in CPD formation in cellular compared to naked samples. The color scale indicates the maximum absolute log$_2$ fold change, such that both positive and negative changes are equally considered. Regions are ordered by gene expression level based on CCLE[48] data (grey bars, right). **b** Strong stimulation as well as inhibition of CPD damage in upstream regions. The bar graph indicates, for each 25 bp window, the strongest stimulatory effect (maximum log$_2$ fold change; green) and strongest inhibitory effect (minimum log$_2$ fold change; red) seen across all promoter regions. Bottom bar graph: lack of net damage change or altered CPD coverage in immediate upstream regions. **c** Per-position cell/naked CPD level change plots for four promoters (remaining are shown in Supplementary Fig. 6). Strong changes in damage (up to -16-fold, i.e. 4 log$_2$ transformed) arise preferably in open regulatory regions indicated by DNaseI hypersensitivity (ENCODE[33]), and large changes are generally lacking in lowly expressed genes such as *TERT*. RPKM, read per kilobase and million reads. Source data are provided as a Source Data file.

individual genomic diPy (cellular samples), to be compared to 0.07 (or 1 CPD per 14 diPy sites) for the same region using available state-of-the-art genome-wide data (202 M CPDs)[10], equivalent to ~970-fold enrichment (Fig. 1g). Capture CPD-seq thus enabled a massive depth of coverage in the assayed human regions, presently unobtainable in non-enriched data, with numerous (up to several hundred) CPDs detected at each individual diPy site.

Principal component analysis of CPD count vectors for all replicates/conditions, separately for each region, revealed clear differences between naked and cellular samples as expected (Supplementary Fig. 2; exemplified by *EGR1* in Fig. 1h). The different cellular conditions (untreated, starved or stimulated) were similar while still forming distinct clusters. In agreement with this, few positions showed statistically altered CPD formation in serum-stimulated compared to starved samples (Supplementary Fig. 3), with *EGR1* exhibiting the strongest effects (three significant base positions at $q < 0.05$; Fig. 1i). Out of four known serum-responsive genes included among the targeted regions, *EGR1* also showed the strongest mRNA induction following serum stimulation in HeLa as determined by qPCR (Supplementary Fig. 4).

### Immediate upstream regulatory regions show strong digital UV footprints

To enable base-level study of CPD changes imposed by DNA binding proteins inside of living cells, we compared CPD formation frequencies in cellular compared to naked replicates at all assayed positions. Fold changes were computed for each nucleotide position while correcting for local ratio bias and differences in CPD frequency distribution across the four diPys in cellular compared to acellular conditions, indicative of differences in basic CPD formation properties (Supplementary Fig. 5, Fig. 1e, Methods).

For an initial overview, we determined the strongest base-specific effect in each 25 bp window across the promoter regions, whether positive (stimulated damage) or negative (inhibited damage). This revealed that genes expressed in HeLa generally showed strongly modulated CPD formation relative to naked DNA immediately upstream of their transcription start sites (TSS), from about -250 to

+1 bp, while such effects were lacking in lowly or non-expressed genes such as *TERT* (Fig. 2a).

Notably, prominent positive as well negative effects were observed, but there was no notable net total change in CPD formation in the immediate upstream regions (Fig. 2b). Many of the changes reached 10-fold or more at specific diPy positions, and strong effects generally coincided with open regulatory DNA as indicated by DNase I hypersensitivity (Fig. 2c, Supplementary Fig. 6). Upstream regulatory DNA sequences thus exhibited prominent and widespread changes in CPD formation in cellular relative to naked DNA, arising at distinct genomic positions and going in both directions, which are obscured in lower-coverage data due to a lack of change in net damage in these regions.

### Distinctive CPD damage signatures at TF binding sites

To investigate the CPD damage footprints at a more detailed level, we focused initially on the promoter of the immediate early gene and transcription factor *EGR1*, known to contain several functional binding sites for SRF (CArG boxes) as well as ETS family transcription factors, which can interact with SRF[19, 27]. Prominent cell/naked differential CPD signals (up to 15.6-fold increase and 8.1-fold decrease) were seen in two upstream regions that coincided with SRF and ETS factor ChIP signals as well as putative binding sites predicted from sequence for these factors (Fig. 3a, regions "1" and "2", both within 440 bp of the TSS). The most significant signals comparing serum-stimulated to starved cells also coincided with the same regions, although the corresponding amplitude changes were weak (Fig. 3a).

The first region ("1") coincided closely with previously identified key *EGR1* regulatory sequences (serum response elements; SREs), which contain several CArG elements known to be bound by SRF as well as ETS factor binding sites[19,28]. We found that the strong cell/naked CPD footprints in this region arose predominantly in these particular binding sites (Fig. 3b). Furthermore, when viewed in a larger context, there was a notable absence of prominent signals in the flanking sequences (Supplementary Fig. 7). Far from being random, CPD signals in *EGR1* were thus structured around known regulatory sites.

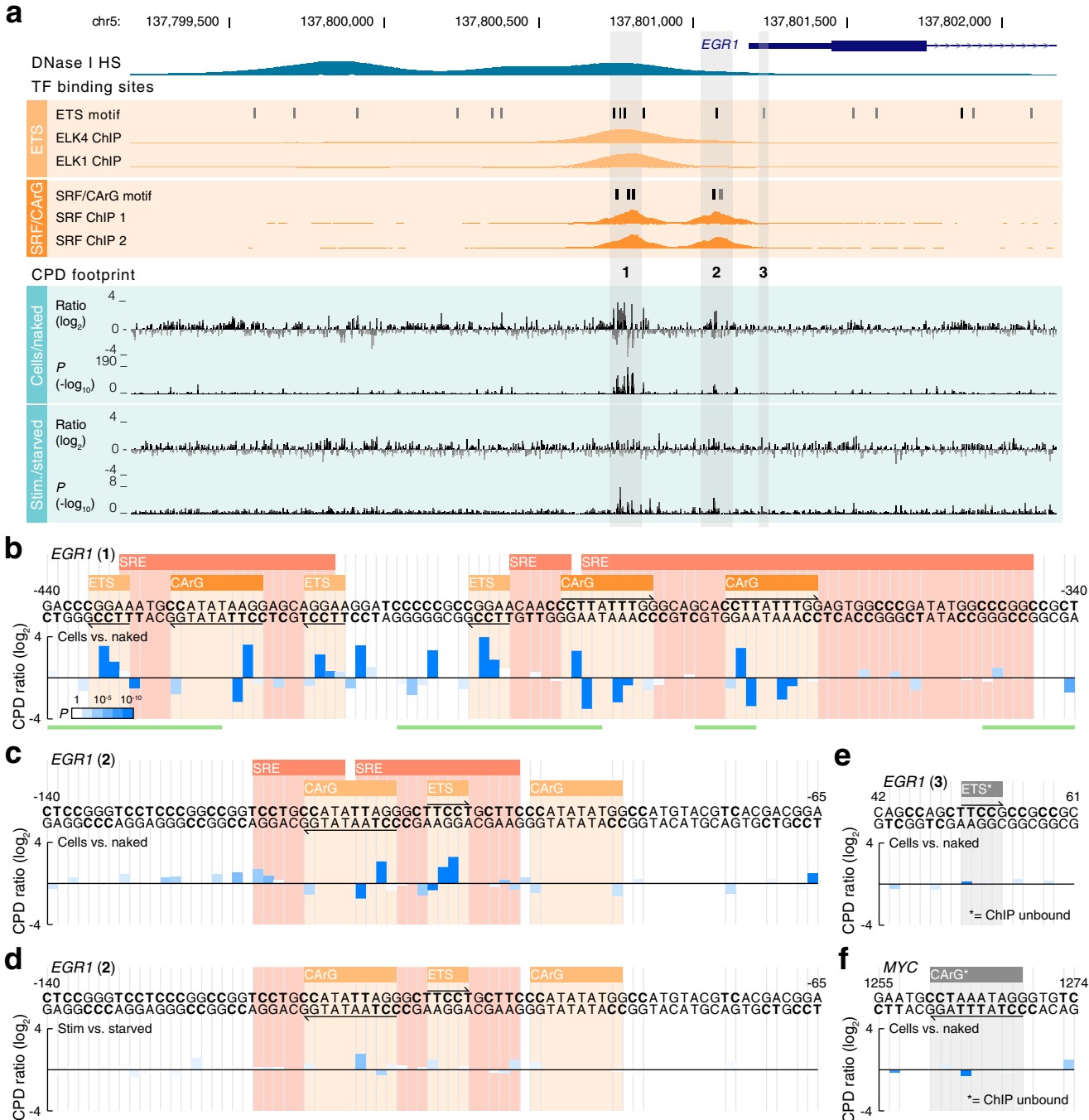

**Fig. 3 | Distinctive CPD damage signatures at SRF and ETS factor binding in the _EGR1_ promoter. a** Overview of the _EGR1_ promoter (−2000 to +1000 bp) showing how Capture CPD-seq UV footprinting signals coincide with regulatory features. Predicted binding sites (sequence motif matches) for SRF (CArG boxes, CC[A/T]$_6$GG) and ETS (TTCC[T/G]) are shown together with ENCODE DNase I and ChIP signals for SRF and ETS (ELK4/ELK1). Per-base CPD damage changes in cellular vs. naked or serum-stimulated vs. starved samples are shown as log$_2$ ratios (+ 4 = 16-fold increase; −4 = 16-fold decrease) together with associated _P_-values (uncorrected from two-sided negative binomial test). CPD-based prediction of bound/unbound status for motif sites is indicated in black/gray, respectively (see Supplementary Fig. 10 for details). **b** Closeup view of region "1" indicated in Fig. 3a. Cell/naked differential CPD signals (up to 15.6-fold) stem primarily from

ChIP-supported SRF and ETS motif matches in previously established serum response elements (SREs)[19,28]. The sites exhibited distinctive repeating CPD damage signatures. Each bar refers to one diPy (only one strand can contain a diPy at each position). **c** Closeup view of a second SRE cluster closer to the _EGR1_ TSS (region "2" in Fig. 3a), containing one informative SRF/CArG element (at −115 bp), and ETS site, and a second CArG lacking diPys in key informative positions. **d** Altered CPD formation in serum-stimulated compared to starved samples at the _EGR1_ −115 bp CArG element (see Supplementary Fig. 8 for details). **e, f** Close-up view of unbound (non-ChIP-supported) ETS and SRF/CArG sites in _EGR1_ (region "3") and _MYC_, respectively, both lacking notable damage modulation signals. DNase I footprints from Vierstra, et al.[31] are indicated as green lines in **b**–**f**. Source data are provided as a Source Data file.

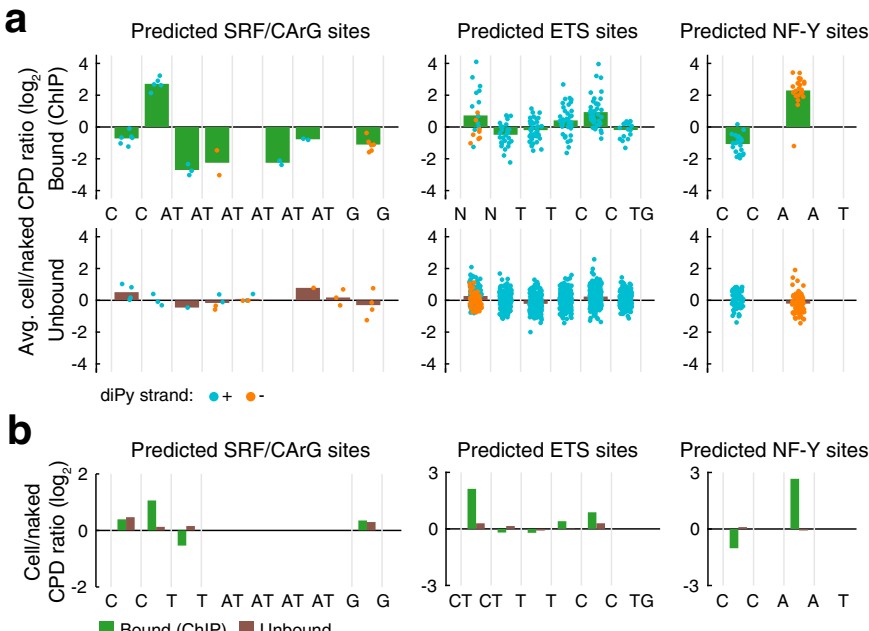

**Fig. 4 | Average CPD damage signatures at bound or unbound predicted TF binding sites. a** Average cell/naked damage ratios (log₂) from Capture CPD-seq at predicted SRF/CArG, ETS and NF-Y sites overlapping with ChIP peak calls (top) or outside of ChIP peaks (bottom). Individual data points are indicated and colored by strand to account for degenerate positions where a diPy may be present on either the forward or the reverse strand, which are structurally distinct situations (a CPD hotspot effect can be seen at the two bases preceding the core TTCC ETS motif when pyrimidines are present at these positions on the forward strand, as described previously[6,14]). Individual predicted sites and ChIP peak calls are shown in Supplementary Fig. 10. **b** Average cell/naked damage ratios at consensus diPy positions for SRF/CArG, ETS and NF-Y in sparse CPD mapping data[10], obtained by summarization of individual CPD detections across thousands of ChIP-supported binding sites and unbound control sites genome-wide. Sequence motifs were selected such that key informative diPy positions are non-degenerate. Source data are provided as a Source Data file.

We next examined the detailed CPD signatures at SRF/CArG and ETS sites in this region ("1"). CArG elements are palindromic with a flexible AT-rich middle part and a consensus leading CC and trailing GG (CC[A/T]₆GG), which is due to SRF binding as a homodimer via its MADS-box DNA binding motif[17,18,20]. We found that, out of three CArG boxes in the region, all had three leading diPys (as in <u>CCTT</u>ATTTGG) and all exhibited a distinctive CPD fingerprint with notable damage stimulation at the second diPy (CT; up to 9.4-fold increase relative naked DNA) and strong inhibition of damage at the third diPy (TT; up to 8.1-fold reduction) (Fig. 3b).

Similarly, three binding sites for ETS factors (TTCC[G/T]), which are known to cooperate with SRF in regulation of early response genes such as *EGR1*[19], exhibited a consistent CPD signature, involving moderately elevated damage at the second (TC) diPy and stronger (up to 15.6-fold) stimulation at the third (CC) diPy (Fig. 3b). ETS factors have been shown to have the capacity to stimulate CPD damage at diPys flanking the core motif on the 5′ side[6,10,22]. In agreement, one of the mapped ETS sites had a leading CC dinucleotide (<u>CC</u>TTCCT; Fig. 3b, second ETS) and exhibited strongly elevated damage (8.7-fold) relative to naked DNA at this position.

Similar distinctive ETS and CArG signatures were seen in the second CArG/ETS-containing region ("2"), which contained a single ETS site and one informative CArG box (Fig. 3c; a second CArG in this region lacked diPy in the critical positions). The damage signature at this CArG element was altered in serum-stimulated compared to starved cells (Fig. 3d), in a manner suggestive of a slight weakening of SRF binding (Supplementary Fig. 8). This is in agreement with recent results showing unchanged or even subtly reduced SRF occupancy at immediate early genes following serum stimulation[29].

Importantly, CArG and ETS sites lacking ChIP support generally exhibited more neutral damage patterns (Fig. 3e, f, Fig. 4a, Supplementary Fig. 9), while summarization of CPD detections across hundreds of ChIP-supported binding sites in sparse

genome-wide CPD mapping data[10] corroborated the SRF and ETS signatures (Fig. 4b).

We similarly investigated CPD damage patterns in the *DPH3/OXNAD1* bidirectional promoter[21], where two distinct cell/naked CPD signals (4.5 and 6.7-fold increases) were seen in a region with two ChIP-supported predicted binding sites for the CCAAT-binding transcription factor NF-Y (nuclear transcription factor Y)[30] (Fig. 5a, region "2"). The signals originated from the two NF-Y sites, and both arose at the same position in the motif (the TT opposite the central AA; Fig. 5b). Consistent CPD inhibition was also seen at the first (CC) diPy in these CCAAT elements. Notably, modulation of CPD damage was generally lacking in NF-Y/CCAAT sites without ChIP support (Fig. 5c, Fig. 4a), and the observed NF-Y signature was in agreement with summary results from sparse genome-wide CPD data[10] (Fig. 4b).

These results support that Capture CPD-seq can reveal structured UV footprints, arising specifically at sites of protein-DNA interaction and involving distinctive damage signatures informative of protein identities. Consequently, we found that a basic classifier applied to cell/naked differential CPD patterns, trained on one or two positive examples, would label predicted SRF, ETS and NF-Y sites as bound or unbound in a manner that closely agreed with information from ChIP-seq (black/gray bars in Fig. 3a and Fig. 5a; Supplementary Fig. 10). Comparison with high-density DNase I cleavage footprints[31] (green bars in Fig. 3b–f and Fig. 5b–d) further highlighted how CPD footprinting provides information that is fundamentally different from DNA accessibility mapping.

### Elevated CPD damage at TF binding sites coincide with cancer mutation hotspots

A second major CPD signal in the *DPH3* upstream region arose at a ChIP-supported ETS binding site shortly upstream of the TSS (Fig. 5a, region "1"). Earlier studies have revealed frequent somatic mutations at

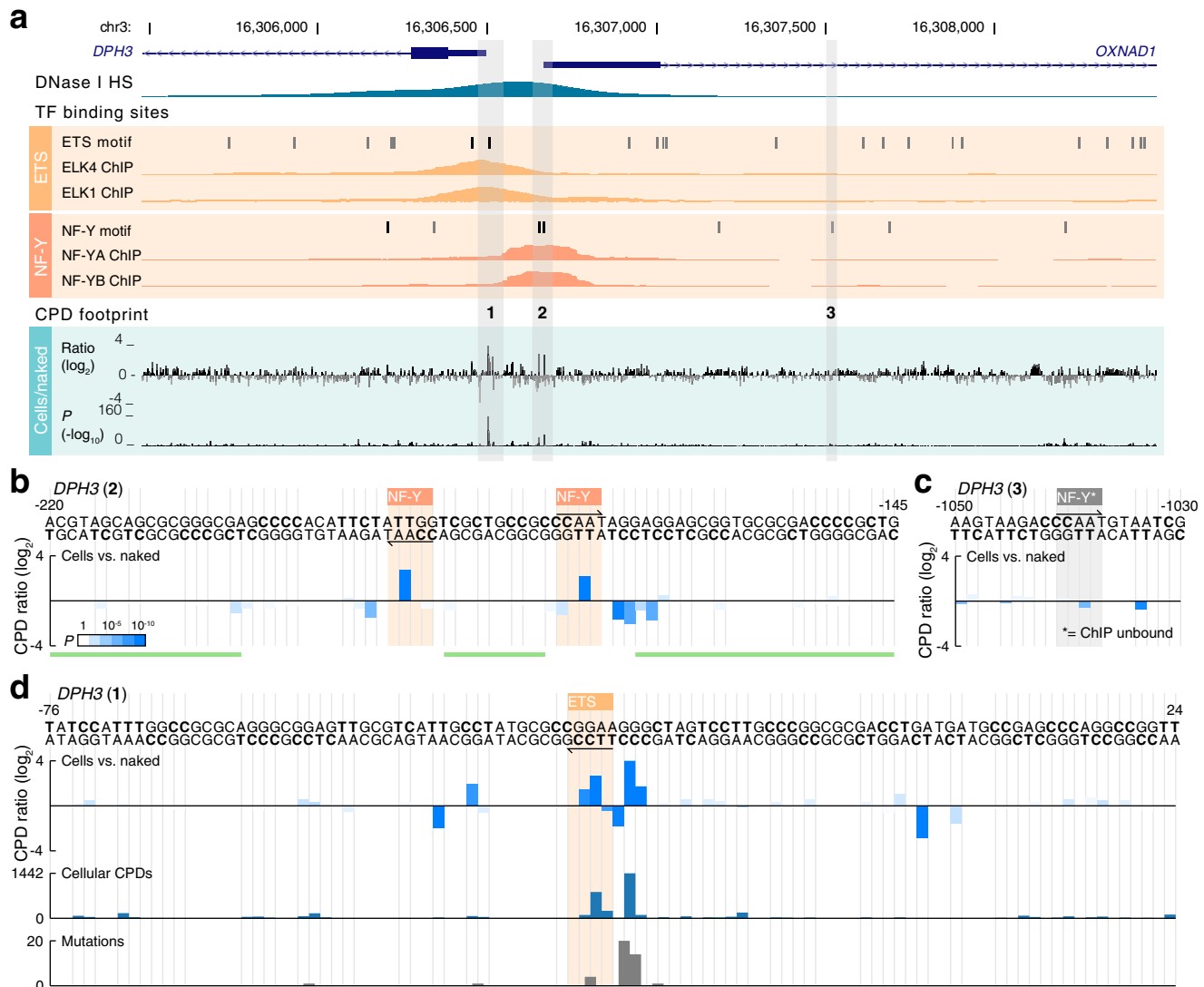

**Fig. 5 | Distinctive CPD damage signatures at NF-Y and ETS binding sites in the *DPH3* promoter. a** Overview of the *DPH3/OXNAD1* bidirectional promoter (−2000 to +1000 bp relative the *DPH3* TSS), including Capture CPD-seq UV footprinting signals and regulatory features. Predicted binding sites (sequence motif matches) for NF-Y (CCAAT) and ETS (TTCC[T/G]) are shown together with ENCODE DNase I and ChIP signals for NF-YA/NF-YB and ETS (ELK4/ELK1). Per-base CPD damage changes in cellular vs. naked samples are shown as log2 ratios (+4 = 16-fold increase; −4 = 16-fold decrease) together with associated *P*-values (uncorrected from two-sided negative binomial test). CPD-based prediction of bound/unbound status for motif sites is indicated in black/gray, respectively (see Supplementary Fig. 10 for

details). **b** Closeup view of region "2" indicated in Fig. 5a. ChIP-supported NF-Y sites in this region exhibit a distinctive CPD damage signature involving strong stimulation of damage (up 6.7-fold) at the center of the motif. Each bar refers to one diPy (only one strand can contain a diPy at each position). **c** Example of an unbound NF-Y site (region "3" in Fig. 5a). **d** Closeup view of a promoter-proximal ETS site in *DPH3* (region "1"). Strongly elevated CPD damage at this site (17.2-fold increase in cell relative naked DNA) coincides at base-level with a prominent somatic mutation hotspot in melanoma (mutation counts based on 221 melanoma whole genome sequences is indicated). DNase I footprints from Vierstra, et al.[31] are indicated as green lines in **b**–**d**. Source data are provided as a Source Data file.

this particular ETS binding site, as well as other ETS sites, in melanoma and other skin cancers[21–23]. Based on sparse genome-wide CPD data, summarized over hundreds of sites, this was suggested to be due to CPD damage being stimulated by ETS proteins[6, 10].

By comparison with somatic mutation data from 221 melanomas, we found that *DPH3* hotspot mutations corresponded at base-level with sharply elevated CPD formation (17.2-fold relative naked DNA) upstream of the core ETS motif (CCTTCCT); indeed the strongest stimulatory effect observed in our dataset (Fig. 5d). Four additional melanoma somatic mutation hotspots (> 3 mutations) were present in the assayed regions: *TERT*, encoding one subunit of telomerase and known to be frequently activated by somatic promoter mutations in cancer[14,24,25], and three ETS-related sites (*RPL13A*, *AP3D1* and *EGR1*). While the latter two ETS hotspots coincided with elevated CPD formation, prominent CPD effects were lacking in *RPL13A* despite high

mutation frequency (Supplementary Fig. 11). Potentially, this can be explained by differences in protein occupancy in HeLa cells compared to melanocytes[14,32,33]. Importantly, notable differential CPD damage was lacking in *TERT*, in agreement with *TERT* promoter mutations arising due to positive selection in skin cancers (Supplementary Fig. 11)[14].

## Discussion

In this study, we show how UV-induced DNA damage can be quantified at base resolution by targeted CPD sequencing (Capture CPD-seq). Using this strategy, we achieved CPD coverage deep enough to detect not only strong hotspots with elevated CPD formation, but also more subtle quantitative effects including reductions at specific positions. This enabled exploration of quantitative CPD damage changes arising at specific human genomic bases due to binding of proteins, at a larger

scale and at higher precision than previously possible[8]. Our results revealed extensive and prominent (up to 17-fold) position-specific changes in UV damage in upstream gene regulatory regions. The changes were positive as well as negative and arose preferably at transcription factor binding sites. Notably, there was no obvious change in total CPD damage burden in DNase-accessible upstream DNA, in agreement with earlier results showing that chromatin state has little net effect on CPD formation[9].

We further show examples of how different proteins can exhibit distinctive damage signatures/footprints that are repeatable in-between binding sites for the same factor, as well as being highly informative in that they encompass elevated as well as inhibited damage at specific positions along their protein-DNA contact interfaces. The information that can be gained from high-throughput quantitative UV damage profiles is thus distinctly different from accessibility-based DNA footprinting[7], and one may envision a collection of damage signatures that, once established, may enable a single experiment to generate occupancy information for multiple different proteins.

Using Capture CPD-seq we were able to reveal base-level agreement between elevated CPD formation in TF binding sites and UV mutation hotspots in melanoma. However, it may be noted that a mutation hotspot in the *RPL13A* promoter, previously shown to be hypersensitive of UV mutagenesis in cultured cells[22], could not be explained by elevated CPD formation, and that several positions exhibiting a high level of CPD damage lacked mutational effects. While this is sometimes explained simply by the non-mutational nature of CPDs formed at TT dinucleotides[34] (such as in NF-Y sites), and while some of the discrepancies may potentially be cell type-related, these results are also likely to reflect a more complex etiology for skin cancer mutation hotspots, with additional contributions from site-specific changes in repair efficacy[35,36] or, putatively, selection during tumor evolution. Nevertheless, although the number of assayed regions was limited, our results show that base-resolution quantification of DNA damage has potential for improving assessment of positive selection in cancer mutation data.

While CPDs can only form at diPys, it may be noted that the purine-pyrimidine complementarity of DNA means that roughly 75% of genomic base pairs are part of a diPy on one strand or the other and thus potentially informative in terms of damage changes. Furthermore, UV light can reach the genome everywhere in living cells and UV damage patterns reflect the state of the DNA at the moment of light exposure. Our results using Capture CPD-seq thus constitute proof of concept for a method with appealing complementary characteristics compared to the existing toolbox. While application of sequencing-based CPD footprinting to complete mammalian genomes still hinges on considerable future reductions in sequencing costs, smaller genomes such as bacteria or even yeast are in principle within reach.

## Methods

### Preparation of DNA samples for capture-CPD-seq

Hela cells were grown in DMEM + 10% FCS + Penicillin/strepto-mycin (GIBCO) and distinct triplicates prepared for three treatment conditions: untreated, starved and stimulated. Starved and stimulated cells were grown in DMEM + 0.5% FCS for 24 h, stimulated cells were stimulated following starvation with DMEM + 20% FCS + 1 μM phorbol myristate acetate (PMA) for 30 min. After each respective treatment, cells were treated with 1000 J/m$^2$ UVC followed by DNA extraction and DNA from untreated cells was isolated as a control, both in duplicates. In addition, nine naked (purified) DNA samples were considered as well as three non-irradiated untreated controls. For naked DNA samples, the DNA was irradiated with the same dose to provide an acellular DNA control sample. DNA was extracted with the Blood mini kit (Qiagen).

### CPD-seq

Purified DNA (12 μg) was sheared to 400 bp using the standard protocol with a Covaris S220 and size selected with SPRI select beads (Life Technologies). The purified product (approx. 5 μg) was subjected to NEBNext end repair and NEBNext dA-tailing modules (NEB). Adapters (ARC141/142) were then ligated to the sheared and repaired ends O/N with NEBNext Quick Ligation module (all oligonucleotide sequences are listed in Supplementary Table 2). DNA was purified with CleanPCR beads (CleanNA) and treated with Terminal Transferase (TdT, NEB) and dideoxy ATP (Roche) for 2 h at 37 degrees. DNA was purified and incubated with 30 units T4 endonuclease V (NEB) at 37 degrees for 2 h, followed by purification and treatment with APE1 (NEB) at 37 degrees for 1.5 h. DNA was purified and treated with rSAP (NEB) 37 degrees 1 h followed by deactivation at 65 degrees for 15 min. DNA was purified, denatured at 95 degrees for 5 min, cooled on ice and ligated with the biotin-tagged adapter ARC143/144 overnight at 16 degrees with NEB-Next quick ligation module. Biotin-tagged DNA fragments were captured with Streptavidin Dynabeads (Invitrogen) and the DNA strand without the biotin label was released with 0.15 M NaOH. A second strand was synthesized using Phusion (Invitrogen) and primer ARC154. The now double-stranded library was purified and amplified with KAPA HiFi HS (Roche) using unique i5 index primers for each replicate (ARC49, ARC156 and ARC157) and unique i7 index primers for each sample (ARC78 and ARC84-91) to add Illumina barcodes and indexes. The libraries were combined into three replicate pools, each containing one of each cellular condition, three naked samples and non-irradiated DNA.

### Capture of human promoter regions with Roche SeqCap EZ

1 μg each of the three replicate pools was hybridized with a custom panel of probes designed using NimbleDesign software (Roche) to cover our promoter panel of interest (Supplementary Table 1). Samples were processed as per the SeqCap EZ HyperCap Workflow (Roche). Briefly, libraries were hybridized with the probe pool first by denaturing at 95 °C for 5 min then incubating at 16 hr at 47 °C. Samples were captured using streptavidin capture beads and a post-capture PCR performed for 8 cycles. Samples were pooled and sequenced using a NextSeq 500 High Output kit (Illumina).

### Capture CPD-seq data mapping and low-level processing

FastQ files were aligned with Bowtie 2 version 2.3.1[37] to hg19, using standard parameters. Duplicate reads were marked with Picard Mark-Duplicates version 2.18.7[38] with the parameter VALIDATION_STRINGENCY = LENIENT, which was essential in removing PCR duplicates arising from the capture protocol. Further analysis was done using a custom processing pipeline, implemented with Bioconductor[39] in R, where CPD positions were extracted as the two bases upstream and on the opposite strand of the first mate in each read pair. Only CPDs detected at diPy sites were considered in the CPD counts and downstream analyses. The processed CPD data was exported in bed format for further processing.

### Statistical analysis of CPD data

For comparisons of cellular vs. naked or serum-stimulated vs. starved samples, double-sided position-specific *P*-values based on the negative binomial distribution, as well as fold changes, were calculated using the Matlab (Mathworks Inc) "rnaseqde" function (with var-iancelink set to constant) while adding one pseudocount to all CPD counts to avoid numerical errors. To correct for a local GC content-related bias in fold change comparing cellular to naked samples, fold changes and *P*-values were calculated based on 250 bp windows (Supplementary Fig. 6). Ratios were further normalized to ensure that each diPy (TT, TC, CT, CC) exhibited an average log$_2$ fold change of zero in each region. This corrects for a skew comparing the four diPys, involving in particular elevated CPD formation at TTs in

naked samples relative to cellular (Fig. 1e), likely arising due to high UV doses and a shift in the CPD formation/reversal equilibrium in-between conditions; an effect that may differentially affect the different diPys[40].

## Integrative analyses and visualization

Preprocessed RNA-seq data was used to estimate expression levels for the included genes/promoters; specifically average FPKM values from HeLa-S3 available from ENCODE under accession codes ENCFF294GCU and ENCFF496CRF. Whole genome somatic mutation calls from 221 melanomas were derived from the Australian Melanoma Genome Project (AMGP)[41], obtained from the International Cancer Genome Consortium's (ICGC) database[42], combined with mutations from the TCGA melanoma cohort[43] called with SAMtools mpileup (options -q1 and -B) and VarScan (strand filter = 1) as described previously[11,23], followed by mapping to the captured regions. Overview panels of select regions (Figs. 3a, 5a and Supplementary Fig. 10) were generated using the UCSC browser [http://genome.ucsc.edu][44]. These overviews included ENCODE DNase I hypersensitivity data from HeLa-S3 (Duke University), and ENCODE ChIP data for ETS factors (ELK1 and ELK4, HeLa-S3), SRF (two replicates from GM12878), and NF-Y (NF-YA and NF-YB, HeLa-S3)[33]. ENCODE uniform peak calls for these datasets were used to classify predicted binding sites as being ChIP-overlapping or outside of ChIP peaks, where ETS sites were required to overlap with both ELK1 and ELK4 while NF-Y sites were required to overlap with both NF-YA and NF-YB. Capture CPD-seq data was exported in wig format for visualization in the UCSC browser. Available high-density DNase I footprints from HeLa-S3 cells (FDR < 0.05) were mapped to the assayed regions and visualized[31]. Putative binding sites were mapped in the assayed regions using the following motifs: ETS, TTCCK; SRF, CCW$_6$GG; and NF-Y, CCAAT.

## CPD-based site occupancy prediction

The bound/unbound status for putative ETS, SRF/CArG and NF-Y TF binding sites was predicted based on CPD data using a two-step approach. An initial sequence analysis to identify binding motif matches was followed by classification applied to the local CPD damage profile, considering only consensus diPy positions in each motif. The motifs and diPy positions used in classification (left-most base on each diPy is underscored) were: ETS, <u>TT</u>CCK (3 positions, all on the positive strand relative to the motif); SRF, <u>CC</u>TW$_5$GG (3 positions of which the last is on the opposite strand relative to the motif); and NF-Y (<u>CC</u>AAT, 2 positions). A basic nearest neighbor classifier was applied to the cell/naked log$_2$ fold changes observed at each motif match, where one or two positive examples with ChIP support were included for each factor (Supplementary Fig. 10), in addition to theoretical unbound/neutral site exhibiting no damage modulation at any of the relevant positions. The predicted class was given as the closest class based on the Euclidian distance.

## Average transcription factor damage signatures from genome-wide CPD data

Previously published sparse whole-genome CPD-seq[10] data from UVC-treated A375 melanoma cells available from GEO using accession number GSE119249 were compared to UVC treated A375 naked DNA CPD-seq from the same study pooled with additional A375 naked DNA CPD-seq generated for this study and available from ENA using accession number PRJEB59730. Average signatures for TFs (ETS, SRF/CArG and NF-Y) were calculated by summarizing CPD detections over bound sites as defined by FactorBook[45], which indicate high-scoring binding sites within ChIP-seq peaks, while matching the following degenerate motifs for each factor: SRF, CCTTW$_4$GG; ETS, YYTTCCK; NF-Y, CCAAT. Sites were additionally required to overlap with DNase I hypersensitive peaks (broad peaks, FDR < 0.01, lowest peak tag density quartile filtered out) from Foreskin Melanocyte primary cells

available from the Roadmap Epigenomics[46] portal using accession code E059. Unbound sites were motif-matching sites, as defined above, that lacked overlaps with ENCODE clustered binding sites for ETS1, ELK4, GABPA, ELF1, NF-Y and SRF (converted from hg38 to hg19 using LiftOver[47]) and ENCODE DNase I peaks. A subset of unbound sites (10× the number of bound sites for each factor) was sampled to speed up calculations. Counts were summarized at each position in the motif, and depth normalization was performed in each condition (cellular or naked) before calculating log$_2$ ratios. Ratios were subjected to correction factors to normalize for a cell vs. naked difference in CPD formation frequency across the four diPy (as seen in Fig. 1e for the Capture CPD-seq data), determined such that global (genome-wide) diPy frequency distributions were the same in the two conditions.

## qPCR assessment of serum stimulation

Hela cells were grown in 6-well plates and subjected to serum starvation and stimulation conditions, as described above for CPD-seq, in triplicate. RNA was extracted using a RNAeasy mini kit (QIAgen) and 1 μg of RNA was converted to cDNA using the Quantitect Reverse Transcription kit (QIAgen). Relative expression of *EGR1*, *FOS*, *JUN*, *MYC*, *STAT3*, *PTGS2* and *GAPDH* was determined using Kickstarter predesigned oligos (Sigma) (Supplementary Table 3) and PowerUP SYBR (Invitrogen) on a Step One Plus Real Time PCR System (Applied Biosystems) with technical duplicates. Expression levels were normalized to *GAPDH* through subtraction of Ct values followed by linearization ($2^{-\Delta Ct}$), and relative levels of stimulation were determined by normalizing the untreated samples to one.

## Reporting summary

Further information on research design is available in the Nature Portfolio Reporting Summary linked to this article.

# Data availability

The Capture CPD-seq data has been deposited in ENA under accession code PRJEB57327. A375 naked DNA whole genome CPD-seq generated for this study has been deposited in ENA under accession code PRJEB59730. The following public datasets were used (converted from hg38 to hg19 using LiftOver[47] where applicable): A375 whole-genome CPD-seq data in GEO, accession GSE119249. HeLa-S3 RNA-seq from ENCODE, accession ENCFF294GCU and ENCFF496CRF. ENCODE HeLa-S3 DNase I peaks from UCSC [https://hgdownload.cse.ucsc.edu/golden Path/hg19/encodeDCC/wgEncodeAwgDnaseUniform/wgEncode AwgDnaseUwdukeHelas3UniPk.narrowPeak.gz]. ENCODE Hela-S3 NFY-A ChIP peaks from UCSC [https://hgdownload.cse.ucsc.edu/ goldenPath/hg19/encodeDCC/wgEncodeAwgTfbsUniform/wg EncodeAwgTfbsSydhHelas3NfyaIggrabUniPk.narrowPeak.gz]. ENCODE Hela-S3 NFY-B ChIP peaks from UCSC [https://hg download.cse.ucsc.edu/goldenPath/hg19/encodeDCC/wgEn codeAwgTfbsUniform/wgEncodeAwgTfbsSydhHelas3NfybIggra bUniPk.narrowPeak.gz]. ENCODE Hela-S3 ELK 1 ChIP peaks from UCSC [https://hgdownload.cse.ucsc.edu/goldenPath/hg19/en codeDCC/wgEncodeAwgTfbsUniform/wgEncodeAwgTfbsSy dhHelas3Elk112771IggrabUniPk.narrowPeak.gz]. ENCODE Hela-S3 ELK 4 ChIP peaks from UCSC [https://hgdownload.cse.ucsc.edu/ goldenPath/hg19/encodeDCC/wgEncodeAwgTfbsUniform/ wgEncodeAwgTfbsSydhHelas3Elk4UcdUniPk.narrowPeak.gz]. ENCODE GM12878 SRF ChIP peaks from UCSC [https://hg download.cse.ucsc.edu/goldenPath/hg19/encodeDCC/wgEn codeAwgTfbsUniform/wgEncodeAwgTfbsHaibGm12878SrfPcr 2xUniPk.narrowPeak.gz]. Hela-S3 DNase I footprints, HeLa-S3-DS24790. Foreskin melanocyte DNase I data from Roadmap Epigenomics, accession E059. Factorbook ETS (ETS1, ELK4, GABPA, ELF1), SRF and NF-Y binding sites from UCSC [http://hgdownload.

soe.ucsc.edu/goldenPath/hg19/database/factorbookMotifPos. txt.gz]. ENCODE clustered transcription factor binding sites from UCSC [http://hgdownload.soe.ucsc.edu/goldenPath/hg38/enc RegTfbsClustered/encRegTfbsClusteredWithCells.hg38.bed.gz]. Cancer Cell Line Encyclopedia HeLa copy number data downloaded from DepMap (20Q2 release) [https://doi.org/10.6084/ m9.figshare.12280541.v4]. WGS-based melanoma mutation calls from the Australian Melanoma Genome Project attained via the ICGC database [https://dcc.icgc.org/projects/MELA-AU]. TCGA WGS melanoma mutation calls were based on alignments downloaded from cgHub (cgHub is no longer available; TCGA WGS data is now accessible through Genomic Data Commons [https:// portal.gdc.cancer.gov]). Researchers need to apply for access to TCGA WGS data to the TCGA Data Access Committee (DAC) via dbGaP (https://dbgap.ncbi.nlm.nih.gov). Source data are provided with this paper.

## Code availability

R and Matlab code written for this study is included as Supplementary Software 1.

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

## Acknowledgements

The results published here are in part based upon data generated by The Cancer Genome Atlas pilot project established by the NCI and NHGRI, as well as ICGC. Information about TCGA and the investigators and institutions who constitute the TCGA research network can be found at [http://cancergenome.nih.gov]. We are most grateful to the patients, investigators, clinicians, technical personnel, and funding bodies who contributed to TCGA and ICGC, thereby making this study possible. The computations were in part performed on resources provided by SNIC through Uppsala Multidisciplinary Center for Advanced Computational Science (UPPMAX) under project b2012108. E.L. is supported by grants from the Knut and Alice Wallenberg Foundation, the Swedish Medical Research Council and the Swedish Cancer Society.

## Author contributions

E.L. and K.E. designed the study. K.E. performed Capture CPD-seq and additional experiments. E.L., K.E., V.K.S., and M.B. analyzed data. E.L. wrote the manuscript with contributions from K.E. and V.S.

## Funding

## Competing interests

The authors declare no competing interests.
