## [Peer Review File · Nature Communications]

Base-resolution UV footprinting by sequencing reveals distinctive damage signatures for DNA-binding proteinsREVIEWER COMMENTS

Reviewer #1 (Remarks to the Author):

The article Base-resolution UV footprinting by sequencing reveals distinctive damage signatures for DNA-binding proteins by Elliott et al describes a new experimental approach (Capture CPD-seq) to detect genomic sites that accumulate recurrent CPDs through capture and sequencing at high depth. They demonstrate that the profile of increase and decrease observed for CPDs (with respect to damage in naked DNA) is reproduced across instances of binding site of a given transcription factor. The manuscript is interesting and well written.

1. In Figure 2a, the scale of log₂ cell/naked ratio should run from negative to positive values, rather than 0-2, to be consistent with Fig. 2b. Moreover, the color scale in 2a may benefit from a change consistent with that of Fig. 2b (red to green).

2. What do bars in the bottom plot of Fig. 2B represent? Are they the average number of CPDs observed at 25 bp windows across selected regions? If that is the case, I would expect that some naked bars be higher than the corresponding cellular bar, since some 25 bp have a preference for damage inhibition. However, at least by eye, this is not apparent. This should be clarified, and the authors might explore an alternative representation for this data.

3. While the mechanisms of stimulation of CPD formation by ETS transcription factors is known and cited by the authors (who contributed to its clarification in the past), the mechanisms potentially underlying damage inhibition are not discussed by the authors. I think they are worth acknowledging and providing the readers with the current knowledge on the subject.

4. I think that the method presented by the authors is indeed interesting for their stated goal of mapping the binding sites of specific families of transcription factors with particular profiles of stimulation and inhibition of CPD formation. Its extension, as stated by the authors in the discussion will depend strongly on the reduction of sequencing costs. Nevertheless, I wonder if the authors may be overlooking other potential applications of the method, such as a higher (almost nucleotide) resolution studies of the interplay between DNA damage and repair, or a more accurate measure of positive selection in mutations in tumorigenesis. I think this would be worth exploring at least in the Discussion section.

Overall, I think the manuscript, in its current state will be very interesting for scientists in the area of genomics. Opening its scope in the directions mentioned above would, I think, help broadening its potential readership.

Reviewer #2 (Remarks to the Author):

The study by Elliott et. al. presents a modification to CPD-seq that allows very high targeted coverage. Authors convincingly showed the utility of this technology by analyzing 21 transcription factor binding sites (TFBS) and 3-5 kb surrounding these sites. The manuscript describes three major biological insights into CPD formation: first, TFBS often have simultaneously protective effect at some sites and stimulating effects at other sites on CPD formation; second, cell conditions like serum starvation have a significant impact on UV-induced lesion formation; third, mutational hotspots in melanoma are associated with CPD hotspots. This is a good proof of principle study and could be published in Nature Communications after the authors will resolve some issues listed below.

The paper is focused on TFBS and shows the effect of TF binding on CPD formation. It would be interesting to expand the scope of the study by analyzing the impact of the cell cycle stage, CpG methylation and hairpins (<https://www.ncbi.nlm.nih.gov/pmc/articles/PMC6731024/>). This addition, while laborious, could significantly broaden the scope of the study. Importantly analysis involving tracks like CpG methylation or hairpins could be done by re-analyzing previously published data. It would be reasonable if the authors decided against additional experiments. It would be interesting to see a version of Fig. 2a and 2b centered around the DHS footprint

(<https://www.ncbi.nlm.nih.gov/pmc/articles/PMC7410829/>) for relevant TF. Footprints offer very high precision and make these panels easier to read.

One of the most important messages of the paper is the different effects of a TF across the motif, where some positions became exposed, and others became protected from the lesions. Statistical presentation of this observation could be significantly strengthened. I would suggest for each region to show variance in, let say, ~ 25 nucleotide sliding window (using window-specific mean). New analysis could be visualized similarly to Fig. 2a. Fig. 2b is a bit confusing, because represents variability between factors and not sites of the same factor.

It is better to move some version of Suppl. Fig. 8a to the main text, because it is an intuitive way to visualize the effect of starvation. You also could aggregate all active TF motive sites to measure the average CPD starved/stimulated ratio. Variance among motif sites could also be a good metric if contrasted with variance in comparable-sized sampled windows. Again, this analysis will sharpen "Importantly, CARG and ETS sites lacking ChIP support generally exhibited near-neutral damage patterns" statement.

Considering limited scope of the study, it is better to be more accurate with language describing the predictive power of CPD hotspots for mutational hotspots in melanoma.

I genuinely appreciate that authors use publicly available data along with their own data to describe site-specific effects.

Minor comments:

I would explain in the main text that PC analyses were applied to the vector of all CPD counts in a region.

Fig. 1d would be better if different conditions for naked DNA were highlighted

QQ plots, e.g., Fig 1i usually done in $-\log_{10}$ transformed coordinates, and while authors imply this transformation, it will be more intuitive to see 0,1,2,3 that corresponds to p-values equal to 10^{-0} , 10^{-1} , 10^{-2} , 10^{-3} . This procedure, will invert axes, and top hits would be in the top right corner
Many figures could benefit from the switch from DHS to DHS footprints.

Point by point response

Manuscript NCOMMS-22-38538-T, “Base-resolution UV footprinting by sequencing reveals distinctive damage signatures for DNA-binding proteins”

We wish to sincerely thank the reviewers for their efforts and insightful comments. As detailed below, we have made a number of changes and additions in response to the feedback from reviewers.

Reviewer comments are shown in black while our responses are shown in blue.

REVIEWER COMMENTS.

Reviewer #1 (Remarks to the Author):

The article Base-resolution UV footprinting by sequencing reveals distinctive damage signatures for DNA-binding proteins by Elliott et al describes a new experimental approach (Capture CPD-seq) to detect genomic sites that accumulate recurrent CPDs through capture and sequencing at high depth. They demonstrate that the profile of increase and decrease observed for CPDs (with respect to damage in naked DNA) is reproduced across instances of binding site of a given transcription factor. The manuscript is interesting and well written.

1. In Figure 2a, the scale of log₂ cell/naked ratio should run from negative to positive values, rather than 0-2, to be consistent with Fig. 2b. Moreover, the color scale in 2a may benefit from a change consistent with that of Fig. 2b (red to green).

This requires some explanation. The heatmap shows the maximum position-specific change in damage for each 25 bp window. It is commonly the case that a single 25 bp window will exhibit both strong stimulatory effects and strong inhibitory effects (as can be seen e.g. within SRF binding sites in **Fig. 3b**).

It is therefore not possible to say whether changes are positive or negative within a window - there will typically be changes in both directions. Hence, absolute values are shown, i.e. the strongest effect whether up or down.

Fig. 2b complements the heatmap in that it shows that there are actually strong changes in both directions. An alternative to the current presentation would be to have two heatmaps (positive and negative effects), but we feel the current figure is better.

We acknowledge the pedagogical challenge and have tried to make things clearer. The heatmap title in **Fig. 2a** was changed to:

“Maximum position-specific cell/naked CPD change per 25 nt window (absolute ratios)”

In results (page 4):

“For an initial overview, we determined the strongest base-specific effect in each 25 bp window across the promoter regions, whether positive (stimulated damage) or negative (inhibited damage).”

Additionally, we have modified the colorscale in the heatmap as some large fold changes were previously out of range.

2. What do bars in the bottom plot of Fig. 2B represent? Are they the average number of CPDs observed at 25 bp windows across selected regions? If that is the case, I would expect that some naked bars be higher than the corresponding cellular bar, since some 25 bp have a preference for damage inhibition. However, at least by eye, this is not apparent. This should be clarified, and the authors might explore an alternative representation for this data.

This relates to the previous question, which highlighted an apparent failure on our side to explain these results in a clear way, and we are grateful to have this pointed out.

The bars show, for each 25 bp window, the total number of CPDs detected across all promoters. There are many prominent changes at many specific positions, but since these changes can be both positive and negative, there is no notable change in the net total damage.

This is an important observation in our opinion: if total damage is measured at lower resolution, there is no obvious change in damage in upstream regions. However, the ability to quantify the damage at individual bases, provided by Capture CPD-seq, reveals a rather massive modulation of damage which would be obscured in low-resolution CPD-seq.

In addition to the clarifications added above, we have now clarified (page 4):

“Notably, prominent positive as well negative effects were observed, and there was no notable net total change in CPD formation in the immediate upstream regions (Fig. 2b).”

In **Fig. 2b** we have changed the labels to “*Max. across all promoters*” and “*Min. across all promoters*”.

3. While the mechanisms of stimulation of CPD formation by ETS transcription factors is known and cited by the authors (who contributed to its clarification in the past), the mechanisms potentially underlying damage inhibition are not discussed by the authors. I think they are worth acknowledging and providing the readers with the current knowledge on the subject.

We thank the reviewer for the suggestions to provide some more in-depth explanation regarding the mechanism underlying modulated CPD damage by proteins.

The main determinant seems to be the distance/alignment between the C5-C5 and C6-C6 carbons that form the cyclobutane structure at the center of the CPD. When these carbons are far apart or when they are immobilized by protein-DNA contacts, damage cannot form easily. Conversely, CPDs can form more easily when these atoms are brought into proximity.

E.g. in 1988, Becker et al (PNAS) interpreted their CPD results from EcoR1 binding to DNA in the context of an x-ray structure of this complex: “Because a high-resolution x-ray crystal structure of the endonuclease-DNA complex is available, we are able to show that strong inhibition of UV photoreactivity at particular bases occurs when the mobility of the photoreactive base is restricted by protein contact. We also demonstrate that kinking of the DNA phosphate backbone by the endonuclease greatly enhances the UV photoreactivity of DNA at the site of the kink.”

We have now added the following explanation to the introduction (page 2):

“These effects are rooted in the positioning and flexibility of DNA bases: UV damage may be inhibited when the mobility of particular pyrimidines is restricted by proteins, while damage may be stimulated at bases that are brought into proximity such that CPD formation is favored⁴⁻⁶.”

4. I think that the method presented by the authors is indeed interesting for their stated goal of mapping the binding sites of specific families of transcription factors with particular profiles of stimulation and inhibition of CPD formation. Its extension, as stated by the authors in the discussion will depend strongly on the reduction of sequencing costs. Nevertheless, I wonder if the authors may be overlooking other potential applications of the method, such as a higher (almost nucleotide) resolution studies of the interplay between DNA damage and repair, or a more accurate measure of positive selection in mutations in tumorigenesis. I think this would be worth exploring at least in the Discussion section.

We agree about this - in particular, improved assessment of positive selection in cancer mutation data warrants to be highlighted more in Discussion given that we show how skin cancer mutation hotspots coincide with CPD hotspots. The Discussion reads (page 7):

“Importantly, using Capture CPD-seq we were able to reveal base-level agreement between elevated CPD formation in TF binding sites and UV mutation hotspots in melanoma.”

And further on page 7:

“Although the number of assayed regions was limited, these results show that base-resolution quantification of DNA damage has potential for improving assessment of positive selection in cancer mutation data.”

Overall, I think the manuscript, in its current state will be very interesting for scientists in the area of genomics. Opening its scope in the directions mentioned above would, I think, help broadening its potential readership.

We wish to thank the reviewer for the effort and the positive feedback.

Reviewer #2 (Remarks to the Author):

The study by Elliott et al. presents a modification to CPD-seq that allows very high targeted coverage. Authors convincingly showed the utility of this technology by analyzing 21 transcription factor binding sites (TFBS) and 3-5 kb surrounding these sites. The manuscript describes three major biological insights into CPD formation: first, TFBS often have simultaneously protective effect at some sites and stimulating effects at other sites on CPD formation; second, cell conditions like serum starvation have a significant impact on UV-induced lesion formation; third, mutational hotspots in melanoma are associated with CPD hotspots. This is a good proof of principle study and could be published in Nature Communications after the authors will resolve some issues listed below.

We thank the reviewer for these appreciative general comments and for the work spent assessing our manuscript.

The paper is focused on TFBS and shows the effect of TF binding on CPD formation. It would be interesting to expand the scope of the study by analyzing the impact of the cell cycle stage, CpG methylation and hairpins (<https://www.ncbi.nlm.nih.gov/pmc/articles/PMC6731024/>). This addition, while laborious, could significantly broaden the scope of the study. Importantly analysis involving tracks like CpG methylation or hairpins could be done by re-analyzing previously published data. It would be reasonable if the authors decided against additional experiments. It would be interesting to see a version of Fig. 2a and 2b centered around the DHS footprint (<https://www.ncbi.nlm.nih.gov/pmc/articles/PMC7410829/>) for relevant TF. Footprints offer very high precision and make these panels easier to read.

We thank the reviewer for these suggestions. It is important to note that, throughout this study we are focusing on comparisons between cellular (i.e. protein-bound) and naked DNA. For features such as methylation and hairpins to have an effect in this comparison, there needs to be a difference in between the two conditions. Below we are discussing these two ideas separately.

Cytosine methylation: while 5mC is known to facilitate CPD formation by UVB light, this is not the case for UVC used in our study (Mitchell et al, *Photochem. Photobiol.* 2000; Rochette et al, *Mutat. Res.* 2009; Lindberg et al, *PNAS* 2019). Additionally, we expect methylation status to be the same in naked vs. cellular DNA, and comparing these two conditions thus makes little sense. In response to this request we instead investigated absolute CPD formation propensity in relation to methylation levels at individual positions using our naked DNA Capture CPD-seq data. However, we failed to see a relationship between the two variables:

Figure 1. Site-specific CPD formation vs. methylation level in Capture CPD-seq. The analysis was done based on publicly available HeLa-S3 methylation data (GSM3633947). Each marker is one position. To compensate for differences in CPD formation propensity due to differing sequence contexts, different tetramers patterns were considered separately, each one normalized relative to the median for each pattern in each captured region.

It should be noted that position-specific biases (e.g. capture or mapping efficacy) will add noise to this analysis, although we believe that an overall correlation should still have been possible to detect. However, to reduce the influence from such effects, we also analyzed CPD levels at individual methylation-compatible positions relative to nearby (within 100 bp) non-methylation-compatible control sites patterns lacking a CG, but found no obvious support for a relationship between methylation and CPD formation:

Boxplot of CPD formation ratios at methylation-compatible positions relative to nearby (<100 bp) related non-methylation-compatible contexts across all captured regions. The analysis was done based on publicly available HeLa-S3 methylation data (GSM3633947) and using naked DNA Capture CPD-seq data.

In the end, the lack of credible evidence for differential CPD formation propensity at highly vs. lowly methylated bases in our data is simply expected given what has previously been established for UVC, and we thus find little reason to include these results in our study.

In regards to hairpins, this is in principle an interesting idea and we have long pondered on the possibility of using Capture CPD-seq to study non-B-DNA structures. In practice there are some challenges - in addition to the question of whether hairpin formation would be differential in cellular vs. naked DNA (all our figures are centered around this

comparison), there is also the issue of hairpin prediction and stability: to pick up a CPD modulation signal from a hairpin (or other secondary structure), the structure needs to be stably present or the signal will be diluted. This does reduce our optimism, but we have nevertheless made an attempt to detect an influence from predicted hairpin structures on CPD formation in the Capture CPD-seq data.

The hairpins in the referenced Buisson et al paper are quite loosely defined, often with short stems, and are also limited to APOBEC-compatible (TpC) loops thus excluding many other CPD-forming loci. Only a single APOBEC hotspot from that study coincided with our regions. In the light of this, we instead predicted putative hairpin structures de novo, settling on a stem length of 6 perfectly complementary bases and a loop size of 3-5 nt to achieve a balance between specificity and sufficient data points. We compared CPD counts in the loop regions of these predicted structures to nearby identical control sequences, for naked DNA as well as cellular DNA:

Analysis of CPD formation in predicted hairpin loops in the captured regions. Hairpins (6 nt stems, 3-5 nt loops) were predicted across all regions and each loop was paired up with a nearby (<200 bp) identical control sequence. CPD counts in predicted loops and proximal controls were compared by means of \log_2 ratios. The analysis was repeated separately for naked and cellular DNA. Positions with low CPD counts (<10) were removed, resulting in 53 and 59 pairs for naked and cellular DNA, respectively. The histograms show the distributions of the paired CPD ratios.

Although there is an expected spread in the ratios, these are centered around zero, no matter if the analysis is done for cellular or naked DNA. In the end, the results are likely to reflect the expectation under the null hypothesis i.e. lack of modulatory effects from non-B-DNA structures. There may simply not be enough sufficiently stable hairpin structures for a clear signal to emerge. We thank the reviewer for encouraging us to investigate this more in detail, even though we did not in the end gain useful insights.

The DNase data seemed a very good suggestion, but upon closer inspection it did not seem to aid in the way we hope in interpreting our results. Many times, the called regions do not line up well even with well-characterized binding sites in early response genes. Called footprints may be shifted relative to binding sites or perhaps fused with nearby signals. This may just be an inherent property of the method: possibly, digestion bias or protein-specific protection properties may lead to footprints not lining up with the actual binding sites in ways that are immediately interpretable, and e.g. for NF- κ B they document an offset in the positioning in the same direction we typically see. While we are certain the data carries useful information, these things add up to not making it very helpful for the reader in our opinion. Example from *EGR1* (well-known serum response elements):

One of the most important messages of the paper is the different effects of a TF across the motif, where some positions became exposed, and others became protected from the lesions. Statistical presentation of this observation could be significantly strengthened. I would suggest for each region to show variance in, let say, ~25 nucleotide sliding window (using window-specific mean). New analysis could be visualized similarly to Fig. 2a. Fig. 2b is a bit confusing, because represents variability between factors and not sites of the same factor.

We fully agree that more systematic statistics on the damage signatures and position-specific stimulatory/inhibitory effects for the different TFs was lacking while being very relevant, including visualization of variability in between bound sites. Likewise, the damage patterns at unbound sites may also be investigated in a more systematic manner, in addition to the individual sites that we highlight (as requested by the reviewer further below).

We have now added an analysis where, for each factor, we split the predicted sites into CHIP-peak-overlapping and non-CHIP-overlapping sets. Sites are thus classified as “bound” or “unbound” using orthogonal data, with the precaution that CHIP does not allow perfect segregation of these two states as CHIP resolution is rather low (peaks are broad). This shows that the distinctive damage signatures described in **Fig. 3-4** arise with a high degree of consistency at bound sites, in particular for NF-Y and SRF/CArG. While the signal is fairly clear also for ETS, overprediction of bound sites is likely to contribute somewhat to a dilution of the average damage signature (site prediction from sequence is fairly promiscuous for this factor and many of the CHIP peaks therefore contain several sites of which some are likely to be false positives). Unbound sites show average patterns that are close to neutral. The analysis is included in the manuscript as **Supplementary Fig. 10** and is also shown below for convenience:

Average CPD damage signatures at bound or unbound predicted SRF, ETS and NF-Y sites. The bargraphs show average cell/naked log₂ fold changes from Capture CPD-seq at predicted binding sites overlapping with ENCODE ChIP peak calls (left column) or outside of ChIP peaks (right column). Individual data points are indicated and colored by strand. This accounts for degenerate positions where a diPy may be present on the forward strand in some individual sites and on the reverse strand in others, which represents two structurally distinct situations. In agreement with the prior literature, a CPD hotspot effect can be seen at the two bases preceding the core TTCC ETS motif specifically when pyrimidines are present at these positions on the forward strand. In the case of ETS, a relatively large number of sequence matches in combination with the limited resolution of the ChIP data likely leads to false positives and a dilution of the average CPD signature.

Fig. 2a/b is meant simply to illustrate that there are in general strong modulation of CPD damage formation (positive as well as negative) in immediate upstream regions, before going into details on the individual sites.

It is better to move some version of Suppl. Fig. 8a to the main text, because it is an intuitive way to visualize the effect of starvation.

There were surprisingly few strong changes following serum stimulation (as stated in the Results, page 4: “few positions showed statistically altered CPD formation in serum stimulated compared to starved samples”) and the changes in the *EGR1* CArG, while significant, were subtle. Therefore, we prefer not to emphasize these results too much, thus preferably including them as supplementary data. However, if the reviewer insists we would naturally include it among the main figures.

You also could aggregate all active TF motive sites to measure the average CPD starved/stimulated ratio. Variance among motif sites could also be a good metric if contrasted with variance in comparable-sized sampled windows. Again, this analysis will sharpen “Importantly, CArG and ETS sites lacking ChIP support generally exhibited near-neutral damage patterns” statement.

We are uncertain about what is meant by windows in this context, but take it to mean the motif-matching regions for the respective factors. We fully agree that the statement about neutral patterns should be backed up by a more systematic analysis of the average damage signature at bound and unbound sites. To this end, we classified our predicted sites as either bound or unbound based on overlap (or lack thereof) with ENCODE peak calls from the same ChIP data used in Fig. 3-4. Please see our reply to point X above which describes these results in detail.

Considering limited scope of the study, it is better to be more accurate with language describing the predictive power of CPD hotspots for mutational hotspots in melanoma.

We agree that this is reasonable. On page 7 in the Discussion, we have now stressed the limited scope in this context:

“Although the number of assayed regions was limited, these results show that base-resolution quantification of DNA damage has potential for improving assessment of positive selection in cancer mutation data.”

I genuinely appreciate that authors use publicly available data along with their own data to describe site-specific effects.

Minor comments:

I would explain in the main text that PC analyses were applied to the vector of all CPD counts in a region.

This is a good suggestion. We have amended the figure legend to contain some more details, and have also amended the main text as follows (page 4):

“Principal component analysis of CPD count vectors for all replicates/conditions, separately for each region, revealed clear differences between naked and cellular samples as expected (Supplementary Fig. 2; exemplified by EGR1 in Fig. 1h).”

Fig. 1d would be better if different conditions for naked DNA were highlighted

The naked samples are all equivalent, since once the DNA is purified it is simply naked HeLa DNA. Hence these are simply 9 replicates of the same condition.

QQ plots, e.g., Fig 1i usually done in $-\log_{10}$ transformed coordinates, and while authors imply this transformation, it will be more intuitive to see 0,1,2,3 that corresponds to p-values equal to 10^{-0} , 10^{-1} , 10^{-2} , 10^{-3} . This procedure, will invert axes, and top hits would be in the top right corner

We thank the reviewer for pointing out an error in this figure. The scale was already logarithmic, and it is thus incorrect to indicate “ $\log_2(P)$ ” on the axes. In addition, the wrong base was indicated as it should be 10, as stated by the reviewer. We have corrected this, and are now showing it in the suggested way, i.e. plotting $-\log_{10}$ transformed values. All plots in **Supplementary Fig. 3** have also been updated accordingly.

Many figures could benefit from the switch from DHS to DHS footprints.

We refer to the discussion above about DHS footprinting data.

REVIEWERS' COMMENTS

Reviewer #1 (Remarks to the Author):

The authors have answered all my previous concerns. I have no further comments

Reviewer #2 (Remarks to the Author):

The authors did an excellent job designing and performing additional tests to address my initial comments. I want to highlight the very high quality of the additional analysis. My only concern is the moderate incorporation of these results into the manuscript and supplementary materials and the disregard for negative results.

Overall, I believe the manuscript should be published in NC after textual issues are addressed. I would like to see all the work done during revision to be published as a part of the manuscript. I am outlining detailed reactions to the revision and ideas on incorporating the results provided in response.

1. I like the additional analysis of CpG methylation. My institution on UV-light dependent CpG mutability is coming from cancer mutagenesis. Moreover, I am not sharing the feeling that it is well-known that methylation does not affect the mutation rate. Also, the review on the topic highlighted the complexity of the interaction between UV-irradiation and CpG methylation [Mechanisms of UV-induced mutations and skin cancer]. It would be great to mention negative results in the text and show figures in the supplement.
2. Similarly, it would be great to mention negative results about hairpins (probably explaining that hairpins are not stable) in the main and show figures in the supplement.
3. The misalignment between footprints and CPD mutability at transcription factor binding sites (TFBS) is a fascinating finding. It is hard to prove that footprints are inaccurate because an orthogonal assessment of TFBS at high resolution is basically impossible. There are few systematic ways to prove that footprints are imperfect using CPD data. One possibility is to compare CPD rate at motifs that overlap both chip-seq + footprint; only chip-seq, but not a footprint or neither. Another option would be to measure the CPD rate at motif as a function of distance from a footprint. For me, this part seems to be the main text panel, but it is for the authors to decide.
4. I suggest merging the new Supplementary Figure 10 with the main text Fig 3. It makes sense to show bound and unbound distributions on the same panel as the authors did for the Fig. 3f,g
5. To this response from authors: "There were surprisingly few strong changes following serum stimulation (as stated in the Results, page 4: "few positions showed statistically altered CPD formation in serum stimulated compared to starved samples") and the changes in the EGR1 CArG, while significant, were subtle. Therefore, we prefer not to emphasize these results too much, thus preferably including them as supplementary data. However, if the reviewer insists we would naturally include it among the main figures."
I would like to see this result as the main figure because of the logic of the text. Although the effect is weak, the result is very interesting and novel.

"The authors did an excellent job designing and performing additional tests to address my initial comments. I want to highlight the very high quality of the additional analysis. My only concern is the moderate incorporation of these results into the manuscript and supplementary materials and the disregard for negative results. Overall, I believe the manuscript should be published in NC after textual issues are addressed. I would like to see all the work done during revision to be published as a part of the manuscript. I am outlining detailed reactions to the revision and ideas on incorporating the results provided in response."

"1. I like the additional analysis of CpG methylation. My institution on UV-light dependent CpG mutability is coming from cancer mutagenesis. Moreover, I am not sharing the feeling that it is well-known that methylation does not affect the mutation rate. Also, the review on the topic highlighted the complexity of the interaction between UV-irradiation and CpG methylation [Mechanisms of UV-induced mutations and skin cancer]. It would be great to mention negative results in the text and show figures in the supplement."

As we explained in our previous response, our CPD data cannot be used for studying this effect as it is based on UVC (which is not present in the sunlight that reaches the earth). It is already known that DNA methylation promotes UV damage induced by UVB - but not UVC. As such we prefer not to include this.

"2. Similarly, it would be great to mention negative results about hairpins (probably explaining that hairpins are not stable) in the main and show figures in the supplement."

We feel this is peripheral to the main story and we don't see a logical place to discuss this in the manuscript: the hairpin predictions are likely to be very noisy and the results are therefore of limited value.

"3. The misalignment between footprints and CPD mutability at transcription factor binding sites (TFBS) is a fascinating finding. It is hard to prove that footprints are inaccurate because an orthogonal assessment of TFBS at high resolution is basically impossible. There are few systematic ways to prove that footprints are imperfect using CPD data. One possibility is to compare CPD rate at motifs that overlap both chip-seq + footprint; only chip-seq, but not a footprint or neither. Another option would be to measure the CPD rate at motif as a function of distance from a footprint. For me, this part seems to be the main text panel, but it is for the authors to decide."

We have added mentioned footprint data to main Figs. 3 and 5 (previously Fig. 4) for all the regions highlighted there (panels 3bcdef and 5bcd), as shown in our previous point-by-point response. It seems hard to prove that the footprints are imperfect and we don't fully understand the proposed analyses, but inclusion of the footprint data in the figures allowed us to highlight that accessibility and UV footprinting provide fundamentally different viewpoints.

"4. I suggest merging the new Supplementary Figure 10 with the main text Fig 3. It makes sense to show bound and unbound distributions on the same panel as the authors did for the Fig. 3f,g"

We have done as suggested. A new main figure has been added (Fig. 4) to avoid Fig. 3 becoming too large, as the next point (5 below) will also add to the size of this figure.

"5. To this response from authors: "There were surprisingly few strong changes following serum stimulation (as stated in the Results, page 4: "few positions showed statistically altered CPD formation in serum stimulated compared to starved samples") and the changes in the EGR1 CArG, while significant, were subtle. Therefore, we prefer not to emphasize these results too much, thus preferably including them as supplementary data. However, if the reviewer insists we would naturally include it among the main figures."

I would like to see this result as the main figure because of the logic of the text. Although the effect is weak, the result is very interesting and novel."

As requested, we have moved parts of Fig. S8 to main Fig. 3 (panel d).

Additional minor changes:

- Added mutation counts to Supplementary Data 1 for all positions
- Removed one of the negative NF-Y site in Fig. 5 (formerly Fig. 4) as this did not add much useful information (and for consistency with Fig. 3 where only one example is shown per factor). Instead we highlighted melanoma mutations and CPD damage in DPH3 over a broader context in Fig. 5, as done in the supplement for the other relevant gene region (Fig. S11)
- Changed the q threshold on row 101 to 0.05 as this is more commonly used, resulting in 3 significant positions
- Various clarifications to Methods
- Minor edits for clarity throughout